# Structures of human dynein in complex with the lissencephaly 1 protein, LIS1

Janice M Reimer[1], Morgan E DeSantis[2], Samara L Reck-Peterson[1,3,4]*, Andres E Leschziner[1,5]*

[1]Department of Cellular and Molecular Medicine, University of California, San Diego, San Diego, United States; [2]Department of Molecular, Cellular and Developmental Biology, University of Michigan, Ann Arbor, United States; [3]Department of Cell and Developmental Biology, University of California, San Diego, La Jolla, United States; [4]Howard Hughes Medical Institute, Chevy Chase, United States; [5]Department of Molecular Biology, University of California, San Diego, La Jolla, United States

**Abstract** The lissencephaly 1 protein, LIS1, is mutated in type-1 lissencephaly and is a key regulator of cytoplasmic dynein-1. At a molecular level, current models propose that LIS1 activates dynein by relieving its autoinhibited form. Previously we reported a 3.1 Å structure of yeast dynein bound to Pac1, the yeast homologue of LIS1, which revealed the details of their interactions (Gillies et al., 2022). Based on this structure, we made mutations that disrupted these interactions and showed that they were required for dynein's function in vivo in yeast. We also used our yeast dynein-Pac1 structure to design mutations in human dynein to probe the role of LIS1 in promoting the assembly of active dynein complexes. These mutations had relatively mild effects on dynein activation, suggesting that there may be differences in how dynein and Pac1/LIS1 interact between yeast and humans. Here, we report cryo-EM structures of human dynein-LIS1 complexes. Our new structures reveal the differences between the yeast and human systems, provide a blueprint to disrupt the human dynein-LIS1 interactions more accurately, and map type-1 lissencephaly disease mutations, as well as mutations in dynein linked to malformations of cortical development/intellectual disability, in the context of the dynein-LIS1 complex.

## Editor's evaluation

This study presents the cryo-EM structure of the dynein regulator Lis1 bound to human dynein providing important insight into how these two proteins interact. The evidence supporting the claims of the authors is convincing overall. The work will be of interest to researchers working with motor proteins and neurodevelopmental disorders as it helps to rationalize how mutations in Lis1 or dynein lead to disease.

## Introduction

Cytoplasmic dynein-1 (dynein here) is a conserved microtubule-based molecular motor. In humans, dynein moves dozens of distinct cargos towards the minus ends of microtubules (*Reck-Peterson et al., 2018*), while in yeast, dynein has a single known role in aligning the mitotic spindle (*Markus et al., 2020*). Active dynein complexes are composed of one or two dimers of dynein, the dynactin complex, and an activating adaptor (*Grotjahn et al., 2018*; *McKenney et al., 2014*; *Schlager et al., 2014*; *Urnavicius et al., 2018*). In cells, dynein dimers are thought to exist primarily in an autoinhibited form (*Amos, 1989*; *Torisawa et al., 2014*; *Zhang et al., 2017*), which is relieved for cargo movement. Recent work has shown that LIS1 has a conserved role in relieving dynein autoinhibition

*For correspondence:
sreckpeterson@ucsd.edu (SLR-P);
aleschziner@ucsd.edu (AEL)

Competing interest: The authors declare that no competing interests exist.

(*Elshenawy et al., 2020*; *Gillies et al., 2022*; *Htet et al., 2020*; *Marzo et al., 2020*; *Qiu et al., 2019*). At a functional level, Lis1 is a dimer of two β-propellers (*Kim et al., 2004*; *Tarricone et al., 2004*). *LIS1* was originally described as the gene mutated in patients with type-1 lissencephaly (*Parrini et al., 2016*; *Reiner et al., 1993*). Later work linked *LIS1* to the dynein pathway (*Sasaki et al., 2000*; *Smith et al., 2000*; *Tai et al., 2002*; *Xiang et al., 1995*). Mutations in the dynein motor-containing heavy chain (*DYNC1H1*) have also been linked to malformations of cortical development (*Lipka et al., 2013*; *Parrini et al., 2016*). Despite the importance of LIS1 in understanding these human diseases, no three-dimensional structures of a human dynein-LIS1 complex have been reported.

Cytoplasmic dynein is a member of the AAA+ (ATPase associated with various cellular activities) family of proteins. Unlike most members of the AAA +family, which are oligomers, the AAA +domains in dynein's 'heavy chain' are fused into a single polypeptide and have diverged over time (*Canty and Yildiz, 2020*). Of dynein's six AAA +domains, four can bind ATP (AAA1-4), and three hydrolyze it (AAA1, AAA3, and AAA4). AAA2 is missing the catalytic glutamic acid needed to hydrolyse ATP, and AAA5 and AAA6 have diverged enough to no longer be able to bind nucleotides (*Schmidt and Carter, 2016*). Dynein's heavy chain can be divided into several elements with specific functions (*Figure 1A*). At its amino-terminus, the 'tail' is responsible for dimerization, and is the site for binding of several accessory subunits. The tail is followed by the 'linker', a mechanical element that undergoes conformational changes, bending at a 'hinge' in response to the nucleotide state of dynein's AAA+ 'ring' to drive movement. Two elements protrude from dynein's ring: the 'stalk', a long antiparallel coiled-coil that protrudes from the ring and ends in dynein's microtubule-binding domain (MTBD), and the 'buttress', a short antiparallel coiled-coil that couples conformational changes in the ring with conformational changes in the MTBD by altering the register between the two alpha helices in the stalk (*Cianfrocco et al., 2015*; *Niekamp et al., 2019*; *Rao et al., 2019*). Dynein's AAA+ ring mainly exists in one of two conformations driven by the nucleotide state of its AAA+ domains: an 'open' conformation coupled to high affinity for the microtubule at the MTBD, and a 'closed' conformation that leads to low affinity for the microtubule (*Schmidt et al., 2015*; *Schmidt et al., 2012*). Dynein's affinity for the microtubule is controlled by conformational changes in the MTBD. Those changes are coupled to the nucleotide state (and therefore conformation) of dynein's motor domain through shifts in the register between the two coiled-coil alpha helices that form dynein's stalk (*Carter et al., 2008*; *Gibbons et al., 2005*; *Redwine et al., 2012*). In dynein's autoinhibited state, called 'Phi' due to its resemblance to the Greek letter, the two heavy chains come together face to face, thus pointing in opposite directions and preventing the motor from engaging microtubules in a manner that would allow it to walk (*Amos, 1989*; *Torisawa et al., 2014*; *Zhang et al., 2017*).

We recently reported a 3.1 Å structure of the yeast dynein-Pac1 complex (*Gillies et al., 2022*). In this structure, dynein is present as a single motor domain bound to two Pac1 β-propellers. This was the first high-resolution structure of a complex between dynein and Pac1 and revealed a number of interactions that earlier, lower resolution maps of the complex had failed to identify (*DeSantis et al., 2017*; *Huang et al., 2012*; *Toropova et al., 2014*). By designing mutants based on this new structure, we showed that binding of Pac1 to dynein, either to its AAA+ ring (site$_{ring}$) or its stalk (site$_{stalk}$), and the interaction between the two Pac1 β-propellers were essential for dynein's function in yeast in vivo (*Gillies et al., 2022*). We also used our model of the yeast dynein-Pac1 complex to generate mutations in human dynein to probe the role of LIS1 in relieving human dynein autoinhibition (*Gillies et al., 2022*). These mutations had relatively mild effects on dynein activation, suggesting that that there may be differences in how dynein and LIS1 interact between yeast and humans that our modeling did not capture.

Here, we set out to determine a high-resolution structure of the human dynein-LIS1 complex. Previously we had obtained 2D class averages from cryo-EM datasets of human dynein in the presence of LIS1 showing that LIS1 binds dynein simultaneously at site$_{stalk}$ and site$_{ring}$, as was the case in yeast (*Htet et al., 2020*). The human and yeast structures appeared very similar at the level of the 2D class averages (*Htet et al., 2020*). We now report 3D cryo-EM structures of human dynein bound to one and two human LIS1 β-propeller domains. We show that there are differences in how human LIS1 binds dynein relative to its yeast counterpart. For both human and yeast, we compare the interactions between dynein and LIS1 at site$_{stalk}$ and site$_{ring}$ and the LIS1-LIS1 interaction. Overall, our work provides a model for how human LIS1 interacts with human dynein. Importantly, our structures also allow us to map missense mutations in type-1 lissencephaly, as well as missense disease mutations in dynein,

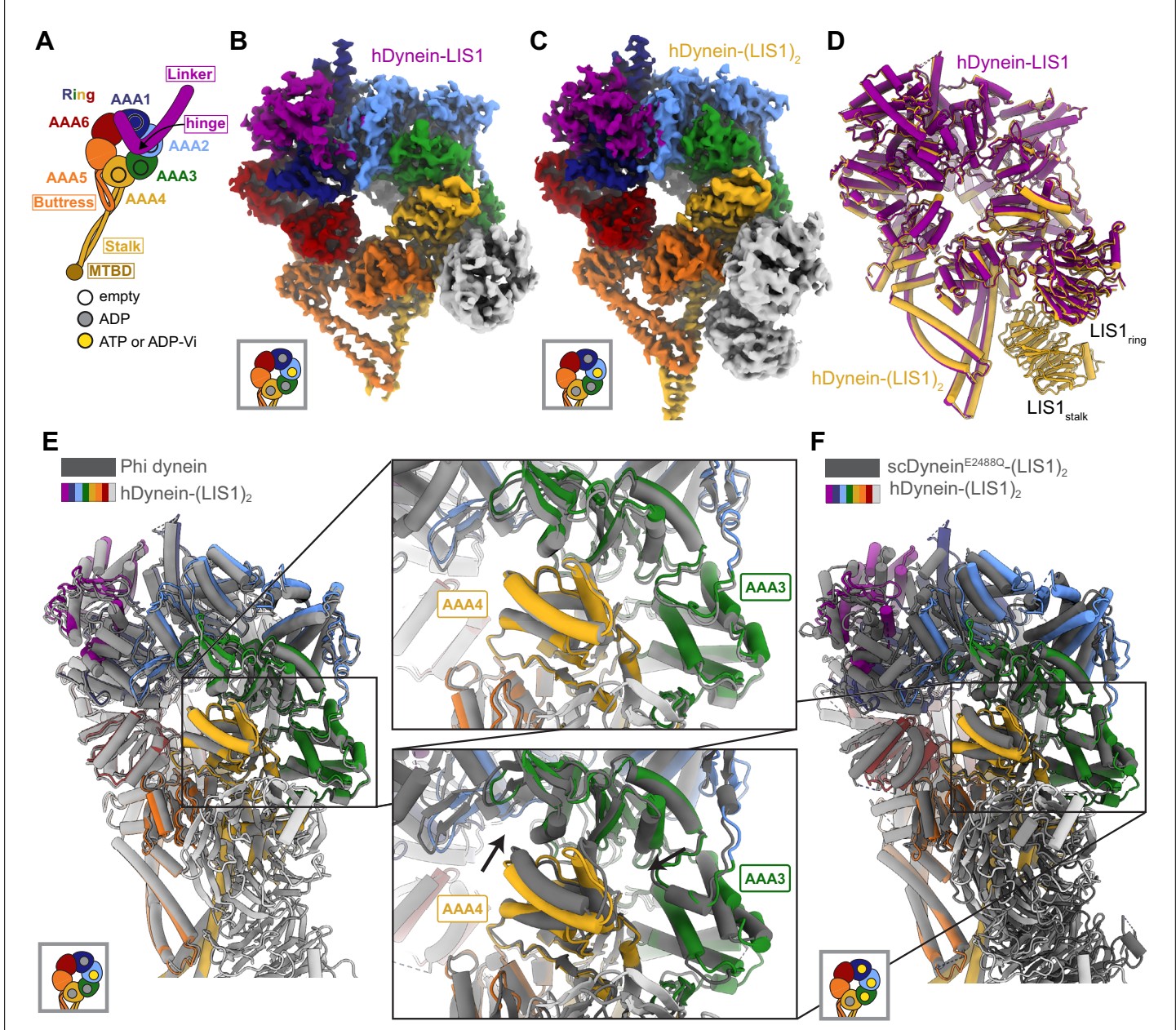

**Figure 1.** Structures of human dynein bound to LIS1. (**A**) Cartoon schematic of dynein showing domain organization. The names of the major structural elements in dynein are indicated inside boxes. MTBD: MicroTubule Binding Domain. The four AAA+ domains that can bind nucleotide are indicated with the black circles. The color coding used throughout the figures to indicate their nucleotide state is shown below dynein's cartoon. (**B, C**) Cryo-EM maps of the motor domain of human dynein bound to (**B**) one (dynein-LIS1) or (**C**) two (dynein-(LIS1)₂) LIS1 β-propellers. (**D**) An overlay of the two human dynein-LIS1 structures solved here. (**E**) An overlay of the human Phi dynein (PDB 5NVU) and the human dynein-(LIS1)₂ structures, aligned on AAA4. The inset shows that the conformation of AAA3 and AAA4 are the same. (**F**) An overlay of the yeast dynein-(Pac1)₂ (carrying a point mutation at E2488Q; PDB 7MGM) and the human dynein-(LIS1)₂ structures, aligned on AAA4. The inset shows there is a slight difference in AAA4 relative to AAA3 between the two structures.

The online version of this article includes the following figure supplement(s) for figure 1:

**Figure supplement 1.** Cryo-EM data processing workflow.

**Figure supplement 2.** Nucleotides bound to dynein.

including those that cause malformations of cortical development and intellectual disability, in the context of dynein's interaction with LIS1.

## Results and discussion

### Structures of human dynein bound to LIS1

We revisited our cryo-EM datasets of human dynein and LIS1 (*Htet et al., 2020*) and, with additional processing, solved the structures of human dynein bound to one and two LIS1 β-propellers to 4.0 Å and 4.1 Å, respectively (*Figure 1B and C*; *Figure 1—figure supplement 1*; *Supplementary file 1*). In both structures, dynein is in the closed ring conformation and the linker domain is disordered before the hinge region. The conformation of dynein is the same regardless of whether LIS1 is bound only at site$_{ring}$ or both at site$_{ring}$ and site$_{stalk}$ (*Figure 1D*). The closed state of the motor domain seen in our structures is the same as that observed in the autoinhibited Phi conformation of human dynein (*Figure 1E*; *Zhang et al., 2017*).

We prepared our samples with ATP and vanadate included in the buffer. Hydrolysis of ATP by dynein in the presence of vanadate leads to the formation of ADP-V$_i$, a post-hydrolysis ADP.P$_i$ analogue. Based on map density, ADP is bound to AAA1 and AAA4, while AAA2 contains either ATP or ADP-V$_i$ (*Figure 1—figure supplement 2*). The nucleotide state of AAA3 is unclear from the density, but we chose to model ADP (*Figure 1—figure supplement 2*) as AAA3 has the same conformation as the ADP-bound AAA3 domain observed in the structure of human Phi dynein (*Figure 1E*), while the

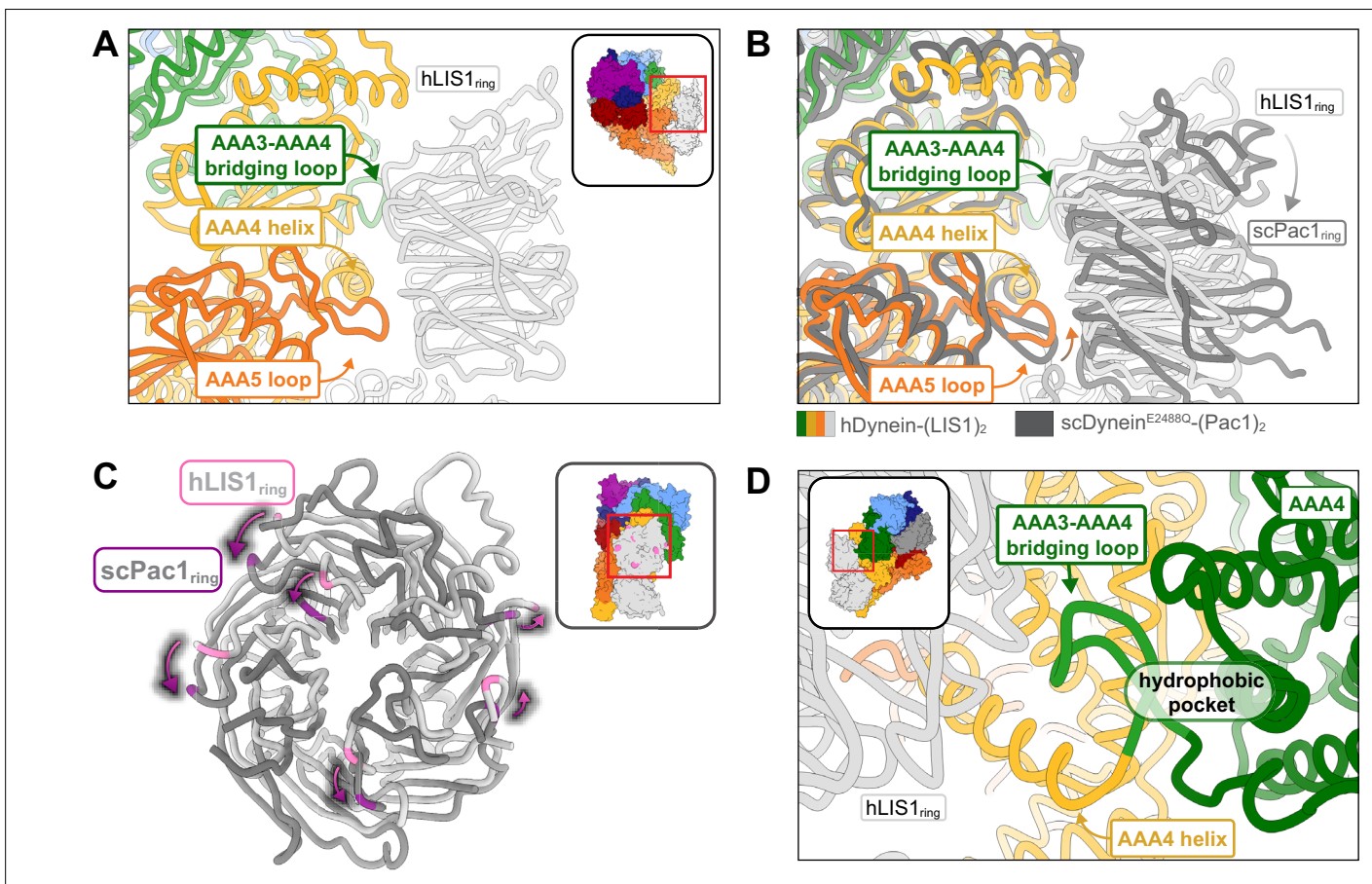

**Figure 2.** Structure of LIS1 binding to dynein at site$_{ring}$. (**A**) LIS1 at site$_{ring}$ interacts with dynein via the AAA3-AAA4 bridging loop, a AAA4 helix and a AAA5 loop. (**B**) An overlay of the human and yeast dynein structures bound to LIS1 /Pac1, aligned on AAA4 (human, light grey; yeast, dark grey). (**C**) LIS1 (light grey) and Pac1 (dark grey) from panel (**B**) are viewed facing the β propeller, with dynein removed for clarity. This panel shows the rotation, highlighted by the purple markers and arrows of LIS1 relative to Pac1 at site$_{ring}$. (**D**) The AAA3-AAA4 bridging loop contacts LIS1 and preserves a hydrophobic pocket in AAA4.

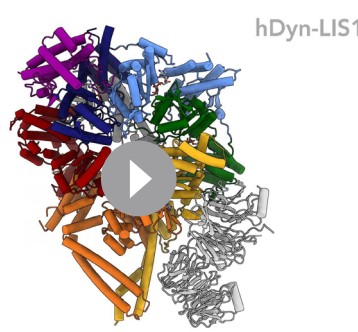

hDyn-LIS1

**Video 1.** Comparison of the human dynein-(LIS1)$_2$ and yeast dynein-(Pac1)$_2$ structures. The video compares the human (dynein-(LIS1)$_2$) and yeast (dynein-(Pac1)$_2$; PDB 7MGM) structures, highlighting some of the major interactions, and the differences in the positions adopted by LIS1/Pac1 at site$_{ring}$ and site$_{stalk}$ in the two systems.

https://elifesciences.org/articles/84302/figures#video1

conformation of AAA3 in yeast dynein (carrying a point mutation, E2488Q, at AAA3 to prevent the hydrolysis of ATP) is different when bound to ATP (*Figure 1F*).

## LIS1 *binding to dynein at site$_{ring}$*

Human LIS1 binds dynein in a manner similar to that of yeast dynein at site$_{ring}$. At site$_{ring}$, the main contact between LIS1 and dynein involves the same AAA4 helix used by yeast dynein, as well as the same AAA5 loop and the loop bridging AAA3-AAA4 (*Figure 2A*; *Video 1*). We previously showed that the AAA4 and AAA5 interactions with yeast Pac1 are important for dynein regulation (*Gillies et al., 2022*). There is a minor rotation in human LIS1$_{ring}$ relative to yeast Pac1$_{ring}$ that causes a slight shift in how LIS1 interacts with dynein (*Figure 2B and C*; *Video 1*). Despite these changes, the interfaces between dynein and LIS1 we saw in our yeast dynein-Pac1 structure are maintained; the AAA5 loop appears to make a small compensating shift to preserve its contact with LIS1 (*Figure 2B*). Additionally, the placement of LIS1 at site$_{ring}$ is the same whether one or two LIS1s are bound to human dynein (*Figure 1D*).

The contact between the AAA3-AAA4 bridging loop and LIS1 (*Figure 2D*) is the pivot point about which the position of LIS1$_{ring}$ rotates between the human and yeast systems (*Video 1*). Given the conservation of this interaction between the two systems, an intriguing possibility is that this contact relays information about the nucleotide state of AAA3, a regulatory site in dynein (*Bhabha et al., 2014*; *DeSantis et al., 2017*; *DeWitt et al., 2015*; *Dutta and Jana, 2019*; *Nicholas et al., 2015*; *Qiu et al., 2021*), to LIS1/Pac1. Although ATP hydrolysis at AAA1 is the main driver of dynein motility, AAA3 has been shown to play a major role in controlling the communication between AAA1 and dynein's MTBD; AAA3:apo or ATP blocks communication, leading to tight microtubule binding, while AAA3:ADP restores dynein's normal mechanochemical cycle (*DeWitt et al., 2015*). The AAA3-AAA4 bridging loop forms a hydrophobic pocket with the small domain of AAA3 (AAA3$_S$), and nucleotide-induced conformational changes in AAA3 may cause this loop and AAA3$_S$ to shift together (*Figure 2D*). Therefore, the bridging loop could act as a tether between LIS1 and AAA3 that allows Lis1 to prevent release of ADP from AAA3. Determining if and how LIS1 modulates the nucleotide state of AAA3 will require structural information on dynein-LIS1 complexes formed in the presence of ATP, as opposed to ATP analogs, allowing dynein to go through its mechanochemical cycle.

## LIS1 *binding to dynein at site$_{stalk}$*

At site$_{stalk}$, LIS1 interacts with dynein at both the CC1 helix in the stalk (the helix leading from dynein's ring to the MTBD) and at a loop in AAA4 (residues 3112–3119) (*Figure 3A*). Human LIS1 is pivoted around the stalk helix relative to yeast Pac1 to a larger extent than at site$_{ring}$ (*Figure 3B* and *Video 1*). We originally used a yeast dynein$^{E3012A\ Q3014A\ N3018A}$ mutant ("dynein$^{EQN}$") (*DeSantis et al., 2017*) to probe the importance of the Pac1-stalk interaction (*Figure 3C*). These mutation sites were chosen based on sequence conservation and low resolution cryo-EM models. Comparing our yeast dynein-Pac1 structure to our new human dynein-LIS1 structure shows that the EQN triad residues are shifted relative to where we had previously modeled them in the human system (*Video 1*), providing an explanation for the modest phenotype we observed when we mutated these residues in human dynein (*Gillies et al., 2022*).

The structure of yeast dynein-Pac1 (*Gillies et al., 2022*) showed that N3018 is the only residue in the EQN triad that forms a hydrogen bond with Pac1, although Q3014 may act to stabilize N3018 by forming a small hydrogen bonding network. R3015 and Q3011 form two additional hydrogen bonds with the backbone of Pac1. Hence, a yeast dynein$^{R3015\ Q3011\ N3018}$ mutant may be better than the original

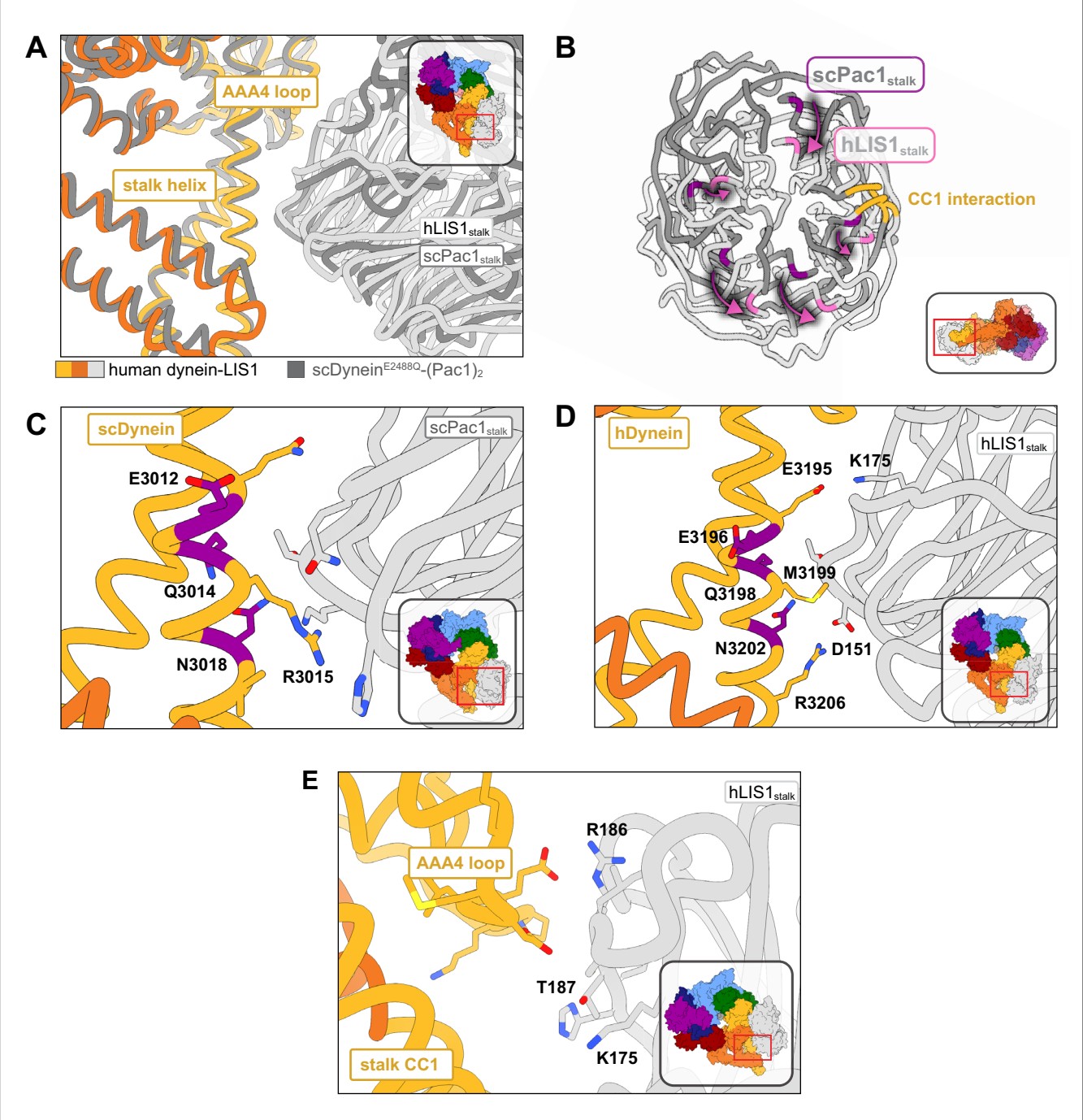

**Figure 3.** Structure of LIS1 binding to dynein at site$_{stalk}$. (**A**) An overlay of human and yeast dynein bound to LIS1/Pac1, aligned on AAA4. (**B**) LIS1 (light grey) and Pac1 (dark grey) from panel (**B**) are viewed facing the β-propeller, with dynein removed for clarity. This panel shows the rotation, highlighted by the purple markers and arrows of LIS1 relative to Pac1 at site$_{stalk}$. The area where LIS1/Pac1 interacts with dynein's CC1 stalk helix is shown in yellow. (**C**) The yeast dynein-Pac1$_{stalk}$ interaction. (**D**) The human dynein-LIS1$_{stalk}$ interaction. (**E**) The AAA4 loop–LIS1$_{stalk}$ interaction.

dynein$^{E2012\ Q3014\ N3018}$ mutant used to disrupt Pac1 regulation at site$_{stalk}$. Similarly, the equivalent EQN triad in human dynein, E3196, Q3198, and N3202, follows the same interaction pattern, where Q3198 interacts with N3202, but only N3202 hydrogen bonds with the backbone of LIS1 (**Figure 3D**). R3206 is in position to form a salt bridge with LIS1 D151, which is made possible by the rotation of LIS1 in the human structure relative to the yeast one. We predict that point mutations M3199A, N3202A, and R3206A in human dynein would disrupt LIS1 regulation at site$_{stalk}$ to a greater extent than dynein$^{E3196}$

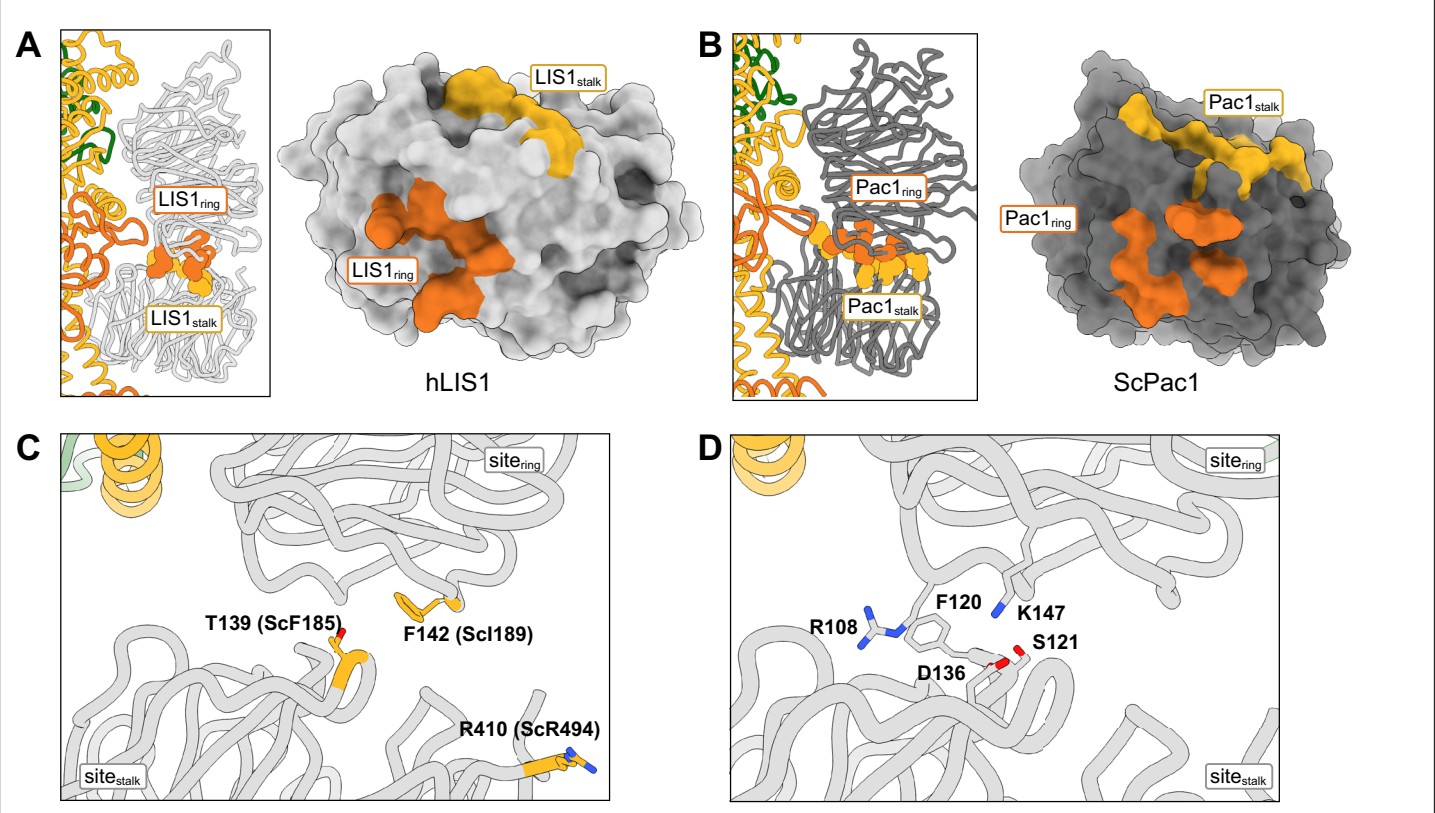

**Figure 4.** Structure of the LIS1-LIS1 interface. (**A, B**) Residues involved in the LIS1-LIS1 interaction are shown in the context of the dynein-(LIS1/Pac1)$_2$ structures and mapped onto a surface representation of LIS1 (**A**) or Pac1 (**B**). Residues involved in the interaction with site$_{ring}$ (LIS1$_{ring}$) are shown in orange and those involved in the interaction with site$_{stalk}$ (LIS1$_{stalk}$) are shown in yellow. (**C**) The human LIS1-LIS1 interaction does not use residues equivalent to those in the yeast Pac1-Pac1 interaction. (**D**) The human LIS1-LIS1 interface.

$^{Q3198\ N3202}$. A septuple mutant we designed in human dynein, dynein$^{K2898A\ E2902G\ E2903S\ E2904G\ E3196A\ Q3198A\ N3202A}$, which comprised the EQN mutations, was still capable of binding LIS1 (*Gillies et al., 2022*).

Previously, we showed that yeast Pac1$^{S248Q}$ acts as a separation-of-function mutant that disrupts the regulation of Pac1 at site$_{stalk}$ without affecting its function at site$_{ring}$ (*Gillies et al., 2022*). In the human structure, the LIS1$_{stalk}$-AAA4 loop interaction is primarily mediated through backbone interactions. Based on sequence alignments, T187 in human LIS1 is homologous to yeast S248; however, in our structure T187 faces away from dynein and is poised to hydrogen bond with K175. To make a separation-of-function mutant in human LIS1, the neighboring residue, R186, which extends towards dynein, may serve as a better mutation candidate in future studies (*Figure 3E*).

## LIS1-LIS1 *interaction*

The biggest difference between the human dynein-LIS1 and yeast dynein-Pac1 complexes is in the LIS1-LIS1/Pac1-Pac1 interaction (*Video 1*). The rotation of LIS1 at site$_{ring}$ and site$_{stalk}$ in the human complex causes the LIS1-LIS1 interface to become significantly smaller, with approximately half the amount of buried surface area (~301 Å$^2$) compared to the yeast Pac1-Pac1 interface (~590 Å$^2$) (*Figure 4A and B*). However, the chemical nature of the interface is also different: while the yeast Pac1-Pac1 interaction is moderately hydrophobic, the human LIS1-LIS1 interface is more electrostatic (*Figure 4C and D*), which may compensate for the smaller surface area.

The yeast Pac1-Pac1 interface mutations (F189D, I189D, R494A) we previously tested (*Gillies et al., 2022*) were designed to disrupt the Pac1-Pac1 interface and are not conserved. Based on structure and sequence alignments, the equivalent residues in human LIS1 (T139, F142, R410) do not participate in the LIS1-LIS1 interface and mutating them would likely not have a disruptive effect (*Figure 4C*). Instead, S121, D136 and K147 may be better candidates to disrupt human LIS1-LIS1 interface (*Figure 4D*).

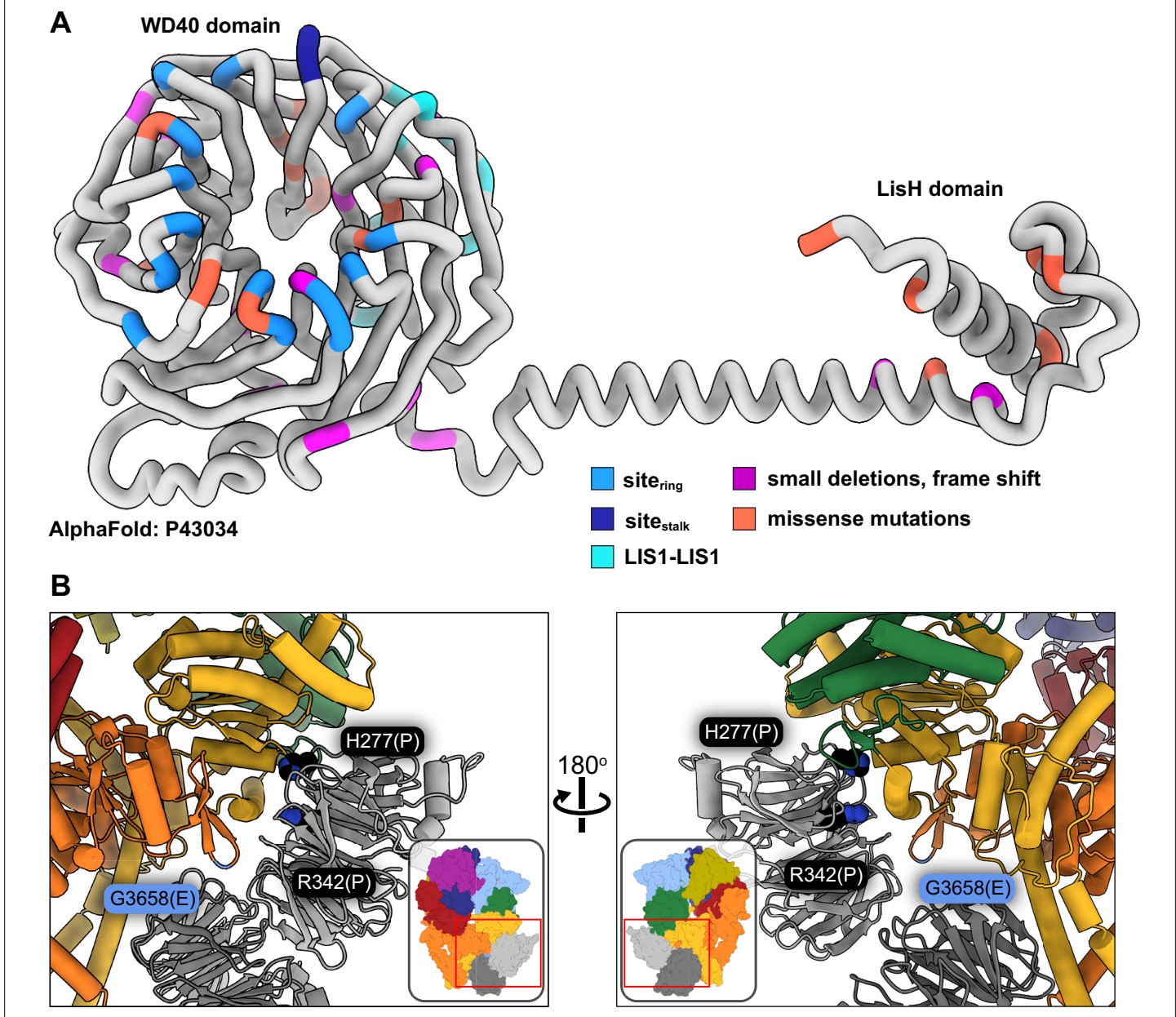

**Figure 5.** Lissencephaly disease causing mutations. (**A**) AlphaFold (*Jumper et al., 2021*; *Senior et al., 2020*) model of full length human LIS1 with residues colored by either interface or lissencephaly mutation. Site$_{ring}$, medium blue; site$_{stalk}$, dark blue; LIS1-LIS1, turquoise; lissencephaly small deletions leading to a frame shift, purple; missense mutations, salmon. (**B**) Two views are shown of disease-linked mutations in dynein located near sites of LIS1 binding. H277P, a lissencephaly mutation, and R342P, a double cortex syndrome mutation, are close to site$_{ring}$. G3658E, associated with intellectual disability, is located at the tip of the AAA5 beta hairpin loop that is part of site$_{ring}$.

## Lissencephaly disease-causing mutations

Lissencephaly is a neurodevelopmental disease caused by mutations in *LIS1* that result in impaired neuronal migration (*Parrini et al., 2016*; *Reiner et al., 1993*). Lissencephaly is a disease of haploinsufficiency and the majority of disease-causing mutations in *LIS1* include large deletions or nonsense mutations that lead to truncated products (*Cardoso et al., 2002*; *Haverfield et al., 2009*; *Lipka et al., 2013*; *Pilz et al., 1998*; *Sapir et al., 1999*). Mutations are located in both the amino-terminal dimerization domain (LisH) and the WD40 domain (*Figure 5A*). Missense mutations are less common. Several missense mutations found in the interior of the WD40 domain are part of the DHSW motifs involved in stabilizing the β-propeller fold and are likely to disrupt the structure of the domain. In *Video 2* we

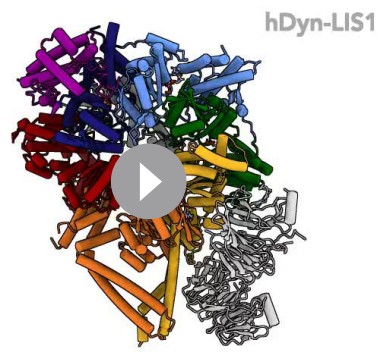

hDyn-LIS1

**Video 2.** Disease mutations in dynein and LIS1. This video shows the location of amino acids in LIS1 mutated in type-1 lissencephaly, and residues in dynein that are mutated in several neurodevelopmental or nondegenerative disorders (Charcot-Marie-Tooth, Spinal Muscular Atrophy, Autism Spectrum Disorders, and Malformations of cortical development/Intellectual disability). We only show residues where we determined that the reported mutation(s) do not have an obvious destabilizing effect based on an inspection of the structure.

https://elifesciences.org/articles/84302/figures#video2

show the location of those missense lissencephaly mutations where a destabilizing effect was not obvious from an inspection of the structure. In addition to the lissencephaly mutations, *Video 2* also shows the location of mutations in LIS1 associated with Miller-Dieker lissencephaly syndrome, subcortical band heterotopia and double cortex syndrome (*Haverfield et al., 2009*; *Pilz et al., 1998*; *Reiner et al., 1993*; *Sapir et al., 1999*, p. 2). *Supplementary file 2* lists all the disease mutations shown in *Video 2*, along with the references first reporting them.

Only two known disease-related missense mutations in LIS1 are near a dynein binding site: both H277P, a lissencephaly mutation, and R342P, a double cortex syndrome mutation, are close to site$_{ring}$ (*Figure 5B*). Although neither amino acid makes a direct contact with dynein, both are involved in hydrogen bonds with nearby residues in the area that comprises the main interface involved in binding to the AAA4 helix at site$_{ring}$. It is likely that local disruption of the structure in the Proline mutants would affect the binding of LIS1 to dynein. H389Y, a subcortical band heterotopia mutation, is located near the LIS1-LIS1 interface. While not part of that interface, H389 makes hydrogen bonds with residues on the same LIS1 β-propeller, including Y137, located in the loop that contains the interface-forming T139.

*Video 2* also shows the location of missense mutations in dynein associated with several neurodevelopmental and neurodegenerative disorders: Charcot-Marie-Tooth, spinal muscular atrophy, autism spectrum disorders, and intellectual disability/malformations of cortical development (*Lipka et al., 2013*; *Reiner et al., 2016*; *Sabblah et al., 2018*; *Weedon et al., 2011*). One mutation is intriguing in terms of LIS1 regulation of dynein: G3658E, which is associated with intellectual disability (*Hertecant et al., 2016*). Although G3658 does not interact with LIS1, it is located at the tip of the AAA5 beta hairpin loop that is part of site$_{ring}$ and is likely involved in the formation of the beta hairpin (*Figure 5B*).

## Conclusions

The cryo-EM structures of human dynein-LIS1 complexes we reported here revealed that while the overall structure of dynein's interaction with LIS1/Pac1 is conserved from yeast to humans, there are important differences in the specifics of the dynein-LIS1/Pac1 and LIS1/Pac1-LIS1/Pac1 interactions. The data and discussion presented here provide a blueprint to better disrupt the human dynein-LIS1 interfaces and to map human disease mutations discovered in the future in the context of the human dynein-LIS1 complex.

## Materials and methods
### Protein purification

The motor domain of human dynein and LIS1 were purified as previously described (*Htet et al., 2020*). In brief, human dynein monomer and LIS1 constructs were expressed in Sf9 cells. Cells were harvested and lysed in dynein-lysis buffer (50 mM HEPES pH 7.4, 100 mM sodium chloride, 1 mM dithiothreitol (DTT), 0.1 mM Mg-ATP, 0.5 mM Pefabloc and 10% (v/v) glycerol) or LIS1-lysis buffer (30 mM HEPES pH 7.4, 50 mM postassium acetate, 2 mM magnesium acetate, 1 mM EGTA, 300 mM potassium chloride, 1 mM DTT, 0.5 mM Pefablock and 10% (v/v) glycerol). Proteins were initially purified using IgG Sepharose 6 Fast Flow beads, following release using TEV protease. Dynein monomer

was further purified using size-exclusion chromatography on a TSKgel G4000SWXL column (TOSOH Bioscience) with GF150 buffer (25 mM HEPES pH 7.4, 150 mM KCl, 1 mM MgCl$_2$, 5 mM DTT and 0.1 mM Mg-ATP). LIS1s final buffer was 10 mM Tris-HCL pH 8.0, 2 mM magnesium acetate, 150 mM potassium acetate, 1 mM EGTA, 1 mM DTT and 10% (v/v) glycerol.

### EM sample preparation

Grids were prepared as previously described (*Htet et al., 2020*). Briefly, UltraAuFoil R1.2/1.3 300 mesh grids (Electron Microscopy Sciences) were glow discharged with 20 mA negative current for 30 s. A 4 µL sample of 3.5 µM dynein monomer, 3.5 uM HaloTag-Lis1 and 2.5 mM ATP-VO$_4$ was applied to the grid and vitrified using a Vitrobot Mark IV robot (FEI) set at 100% humidity and 4 °C.

### EM data collection

Data collection was performed as previously described (*Htet et al., 2020*). Three datasets were collected and initially processed separately. Briefly, each dataset was processed in cryoSPARC using the patch motion correction and patch CTF extraction jobs to align micrographs and perform CTF estimation, respectively. Micrographs with a CTF estimation of >5 Å were discarded. Dose weighted images were used for particle picking using the crYOLO training model generated in *Gillies et al., 2022*; *Wagner et al., 2019*. Particles were extracted in Relion 3.0 (\*\*\*ref\*\*\*) with a 1.16 Å/pixel. Several rounds of 2D classification were carried out in cryoSPARC to remove bad particles. Particles belonging to good 2D class averages in datasets 1 and 2 were combined and ab initio reconstruction was carried out. Particles belonging to dynein were combined and heterogeneous refinement was carried out to separate intact dynein from partially unfolded dynein. Another round of heterogenous refinement was carried out that included the good particles from Dataset 3. Particles were separated into 1 Lis1 and 2 Lis1 classes, and each resulting map was used in nonuniform refinement (*Punjani et al., 2020*). The final resolution of human dynein-Lis1 and human dynein-(Lis1)$_2$ was 4.0 Å and 4.1 Å, respectively.

We note that the overall resolution of our structures was limited due to preferred orientation. These datasets were collected on open hole grids before we began using streptavidin affinity grids, which helped reduce this problem in our most recent structure of yeast dynein-Pac1.

### Model building

The structure of Phi dynein (PDB 5 NUG) and the AlphaFold model of human Lis1 (model P43034) were used as initial models for the human dynein-Lis1 structure and fit into the map using UCSF ChimeraX (*Pettersen et al., 2021*). Refinement of the model was carried out using a combination of Phenix Real Space Refine (*Afonine et al., 2018*) and Rosetta Relax (ver 3.13). Parts of the model were manually rebuilt using COOT (*Emsley et al., 2010*). Following completion of the human dynein-Lis1 model, it was used as a starting model for dynein-(Lis1)$_2$ where Lis1$_{ring}$ was duplicated and fit into the position at site$_{stalk}$ using UCSF ChimeraX. Refinement proceeded using the same method as for dynein-Lis1.

## Acknowledgements

We thank Eva Karasmanis and Agnieszka Kendrick for editing the manuscript. We thank our funding sources: JMR was a Merck Fellow of the Damon Runyon Cancer Research Foundation, DRG-2370–19; MED was a Jane Coffin Childs Postdoctoral Fellow; SRP's lab is supported by the Howard Hughes Medical Institute and NIH R35 GM141825; AL's lab was supported by NIH R01 GM107214 and is now supported by R35 GM145296. We thank Zaw Min Htet and Gillies for help with protein purification, and Richard W Baker for help with cryo-EM data collection. We also thank the UC San Diego Cryo-EM facility, and the UC San Diego Physics Computing Facility for IT support.

## Additional information

### Funding

| Funder | Grant reference number | Author |
|---|---|---|
| Damon Runyon Cancer Research Foundation | DRG-2370-19 | Janice M Reimer |
| Jane Coffin Childs Memorial Fund for Medical Research | | Morgan E DeSantis |
| Howard Hughes Medical Institute | | Samara L Reck-Peterson |
| National Institute of General Medical Sciences | R35 GM141825 | Samara L Reck-Peterson |
| National Institute of General Medical Sciences | R01 GM107214 | Andres E Leschziner |
| National Institute of General Medical Sciences | R35 GM145296 | Andres E Leschziner |

The funders had no role in study design, data collection and interpretation, or the decision to submit the work for publication.

### Author contributions

Janice M Reimer, Conceptualization, Formal analysis, Validation, Investigation, Visualization, Methodology, Writing - original draft, Writing - review and editing; Morgan E DeSantis, Conceptualization, Investigation, Writing - review and editing; Samara L Reck-Peterson, Andres E Leschziner, Conceptualization, Supervision, Funding acquisition, Writing - original draft, Project administration, Writing - review and editing

### Author ORCIDs

Morgan E DeSantis http://orcid.org/0000-0002-4096-8548
Samara L Reck-Peterson http://orcid.org/0000-0002-1553-465X
Andres E Leschziner http://orcid.org/0000-0002-7732-7023

### Decision letter and Author response

Decision letter https://doi.org/10.7554/eLife.84302.sa1
Author response https://doi.org/10.7554/eLife.84302.sa2

---

# Additional files

### Supplementary files

• Supplementary file 1. Cryo-EM data information and model validation. CryoEM data collection parameters, reconstruction information and model refinement statistics for the structures of human dynein bound to one and two LIS1 β-propellers.

• Supplementary file 2. Disease mutations in dynein and LIS1 shown in Video 2. This table lists the mutations in dynein and LIS1 that are associated with different disease and are shown in Video 2, along with the references reporting them. Video 2 shows (and this Table lists) only residues where we determined that the reported mutation(s) do not have an obvious destabilizing effect based on an inspection of the structure. The color coding in the table corresponds to that used in Video 2.

• MDAR checklist

### Data availability

Cryo-EM maps and resulting models have been deposited in the EM Data Bank (maps) and PDB (models). Raw micrographs have been deposited in EMPIAR. Accession numbers are listed in Supplementary file 1.

The following datasets were generated:

| Author(s) | Year | Dataset title | Dataset URL | Database and Identifier |
|---|---|---|---|---|
| Reimer JM, DeSantis ME, Reck-Peterson SL, Leschziner AE | 2022 | Structures of human dynein in complex with the lissencephaly 1 protein, LIS1 | https://www.ebi.ac.uk/emdb/search/EMD-27783 | ArrayExpress, EMD-27783 |
| Reimer JM, DeSantis ME, Reck-Peterson SL, Leschziner AE | 2022 | Structures of human dynein in complex with the lissencephaly 1 protein, LIS1 | https://www.ebi.ac.uk/emdb/search/EMD-27782 | ArrayExpress, EMD-27782 |
| Reimer JM, DeSantis ME, Reck-Peterson SL, Leschziner AE | 2022 | Structures of human dynein in complex with the lissencephaly 1 protein, LIS1 | https://www.rcsb.org/structure/8DYV | RCSB Protein Data Bank, 8DYV |
| Reimer JM, DeSantis ME, Reck-Peterson SL, Leschziner AE | 2022 | Structures of human dynein in complex with the lissencephaly 1 protein, LIS1 | https://www.rcsb.org/structure/8DYU | RCSB Protein Data Bank, 8DYU |
| Reimer JM, DeSantis ME, Reck-Peterson SL, Leschziner AE | 2023 | Structures of human dynein in complex with the lissencephaly 1 protein, LIS1 | https://www.ebi.ac.uk/empiar/EMPIAR-11373/ | EMPIAR, EMPIAR-11373 |

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
