## [Editor Report]

This study presents the cryo-EM structure of the dynein regulator Lis1 bound to human dynein providing important insight into how these two proteins interact. The evidence supporting the claims of the authors is convincing overall. The work will be of interest to researchers working with motor proteins and neurodevelopmental disorders as it helps to rationalize how mutations in Lis1 or dynein lead to disease.

---

## [Decision Letter]

**Decision letter after peer review:**

Thank you for submitting your article "Structures of human dynein in complex with the lissencephaly 1 protein, Lis1" for consideration by *eLife*. Your article has been reviewed by 3 peer reviewers, and the evaluation has been overseen by a Reviewing Editor and Anna Akhmanova as the Senior Editor. The following individuals involved in review of your submission have agreed to reveal their identity: Xin Xiang (Reviewer #1); Deanna Smith (Reviewer #2); Kashish Singh (Reviewer #3).

Essential revisions:

1) Please address specific points 1 and 2 of Reviewer 3. Specifically, please try to improve the resolution of the reported structures and please clarify the nucleotide state of AAA1.

2) Please include a detailed Supplementary table specifying the mutations you have mapped with appropriate references.

3) Please consider the other suggestions of the reviewers.

*Reviewer #1 (Recommendations for the authors):*

The authors have previously reported a 3.1Å structure of yeast dynein bound to Pac1 (budding yeast LIS1) (Gillies et al., 2022, *eLife*). However, mutations they designed using the yeast dynein-PAC1 structure had mild effects on human dynein activation. Here they reported cryo-EM structures of human dynein-LIS1 complexes. While LIS1 and Pac1 binds to roughly the same sites of the dynein motor domains at the level of the 2D class averages, their current 3D cryo-EM structures of human dynein bound to one and two human LIS1 β-propeller domains (4.0 Å and 4.1 Å resolution respectively) have revealed interesting similarities and differences in the interaction sites. Although this work does not contain functional analyses on the involved amino acids in interaction interfaces (unlike the Gillies et al., 2022 paper), the authors have provided detailed information that could be followed to perform such analyses in the future. In addition, they have also provided the locations of missense mutations of LIS1 and dynein that cause lissencephaly and other human brain developmental or neurodegenerative disorders in the context of the human dynein-LIS1 structure. Overall, I found the work very interesting and well presented. This first detailed structural analysis on human dynein-LIS1 interaction is likely to have a high impact in the field.

I have the following suggestions.

1. I really enjoyed the two videos, especially Video 2, which shows lissencephaly disease mutations, as well as mutations causing brain development disorders and intellectual disability, in the context of dynein-LIS1 interaction. If space permits, it would be helpful to have a supplementary table listing each of these mutations and specific references. The authors did describe the mutations in the final section of the text with references, but multiple references are cited together and the reader would need to go through them to find the correct reference. For example, I was curious why the R342P mutation (line 191) in LIS1 causes double cortex (as the lis1 gene is not X-chromosome-linked), and is the original reference Jamuar…… Walsh (2014)?

2. Line 38, add Marzo et al. 2020. Line 52-59, it may be better to also include a few new references after the 2015 review, for example, Rao et al. 2019; Niekamp et al., 2019.

Line 123, add Bhabha et al., 2014.

3. It is interesting that the closed state of the motor domain seen by the authors in their structures is similarly observed in the Phi dynein, which contains ADP in AAA3 (Zhang et al., 2017). Since the yeast AAA3 walker B dynein (contains ATP) has Pac1 bound to the ring and stalk, I wonder if this represents a difference between the two systems and is worthy of a brief discussion. Also, since the phi structure was previously predicted to be incompatible for LIS1 binding (Htet et al., 2020; Marzo et al., 2020), a brief discussion may help the understanding of the new data too. It would also be better to point out in Figure 1 or in Materials and methods that the motor domain of human dynein rather than full-length dynein is used for this study.

4. Line 148-149 "…point mutations at M3199A, N3202A, and R3206A", remove "A" after the residues as you are talking about the position, unless you remove "at".

5. Line 155-158 "…. in human LIS1 is homologous to yeast S248; however, in our structure T187 faces away from dynein and is poised to hydrogen bond with K175. To make a separation-of-function mutant in human LIS1, the neighboring residue, R186, which extends towards dynein, may serve as a better mutation candidate in future studies (Figure 3E)." It may be better to add a sentence or two here to say what the predicted effect of such a separation-of-function mutation would be on human dynein activation based on the previous data from the budding yeast.

6. Although I do not think functional tests are essential for publication of this structural paper, it would be nice to test if some mutations (designed according to the new structure) affect LIS1's ability to bind dynein or enhance the speed of dynein motility in vitro. Cell-based tests are even better, as the seperation-of-function mutation may affect dynein-mediated cargo transport but not dynein accumulation at the microtubule plus end (Splinter et al., 2012). However, functional assays in humans cells are much harder than those in budding yeast, and some functional tests may need to be done after the publication of this new structure.

*Reviewer #2 (Recommendations for the authors):*

I recommend determining whether the K147A mutation in fact disrupts the interaction between b-propellars in human LIS1 and if that prevents dynein stimulation.

---

## [Author Response]

Essential revisions:1) Please address specific points 1 and 2 of Reviewer 3. Specifically, please try to improve the resolution of the reported structures and please clarify the nucleotide state of AAA1.

We have done additional processing of our data, following the reviewer’s suggestions. Although we were able to increase the resolution by a modest 0.3Å, the new maps have not resolved any of the ambiguities of our original ones. See our response to reviewer #3 for more information.

We are including a new Reviewer-Figure-1 showing the lack of improvement in the density around the nucleotide-binding site in AAA1 despite the nominal increase in overall resolution.

2) Please include a detailed Supplementary table specifying the mutations you have mapped with appropriate references.

The revised manuscript includes a new “Supplemental Table 2: Disease mutations in dynein and LIS1 shown in Video 2”.

3) Please consider the other suggestions of the reviewers.

All suggestions are addressed below.

Reviewer #1 (Recommendations for the authors):The authors have previously reported a 3.1Å structure of yeast dynein bound to Pac1 (budding yeast LIS1) (Gillies et al., 2022, eLife). However, mutations they designed using the yeast dynein-PAC1 structure had mild effects on human dynein activation. Here they reported cryo-EM structures of human dynein-LIS1 complexes. While LIS1 and Pac1 binds to roughly the same sites of the dynein motor domains at the level of the 2D class averages, their current 3D cryo-EM structures of human dynein bound to one and two human LIS1 β-propeller domains (4.0 Å and 4.1 Å resolution respectively) have revealed interesting similarities and differences in the interaction sites. Although this work does not contain functional analyses on the involved amino acids in interaction interfaces (unlike the Gillies et al., 2022 paper), the authors have provided detailed information that could be followed to perform such analyses in the future. In addition, they have also provided the locations of missense mutations of LIS1 and dynein that cause lissencephaly and other human brain developmental or neurodegenerative disorders in the context of the human dynein-LIS1 structure. Overall, I found the work very interesting and well presented. This first detailed structural analysis on human dynein-LIS1 interaction is likely to have a high impact in the field.I have the following suggestions.1. I really enjoyed the two videos, especially Video 2, which shows lissencephaly disease mutations, as well as mutations causing brain development disorders and intellectual disability, in the context of dynein-LIS1 interaction. If space permits, it would be helpful to have a supplementary table listing each of these mutations and specific references. The authors did describe the mutations in the final section of the text with references, but multiple references are cited together and the reader would need to go through them to find the correct reference. For example, I was curious why the R342P mutation (line 191) in LIS1 causes double cortex (as the lis1 gene is not X-chromosome-linked), and is the original reference Jamuar…… Walsh (2014)?

We thank the reviewer for this suggestion. We have now added Supplemental Table 2, which contains a list of all the mutations shown in Video 2, along with references. We color-coded the table to match the color-coding of Video 2.

2. Line 38, add Marzo et al. 2020. Line 52-59, it may be better to also include a few new references after the 2015 review, for example, Rao et al. 2019; Niekamp et al., 2019.

References added as suggested.

Line 123, add Bhabha et al., 2014.

Reference added.

3. It is interesting that the closed state of the motor domain seen by the authors in their structures is similarly observed in the Phi dynein, which contains ADP in AAA3 (Zhang et al., 2017). Since the yeast AAA3 walker B dynein (contains ATP) has Pac1 bound to the ring and stalk, I wonder if this represents a difference between the two systems and is worthy of a brief discussion. Also, since the phi structure was previously predicted to be incompatible for LIS1 binding (Htet et al., 2020; Marzo et al., 2020), a brief discussion may help the understanding of the new data too. It would also be better to point out in Figure 1 or in Materials and methods that the motor domain of human dynein rather than full-length dynein is used for this study.

Regarding the comparison of the yeast and human structures bound to Pac1/LIS1, we think that additional cryo-EM will need to be done to capture the full range of conformations and nucleotides in dynein that are compatible with Pac1/LIS1 binding. This would allow a better comparison of the yeast and human systems. Time-resolved cryo-EM after the addition of ATP might be a way to determine this. Ongoing time-resolved work in the lab using the yeast system has already shown that the closed conformation is compatible with several different permutations of nucleotide states in dynein’s motor domain. Future work should help us determine whether the same is true in the human system, but this is beyond the scope of the current manuscript.

The Phi structure is sterically incompatible with Pac1/LIS1 binding because a Pac1/LIS1 bound to site_ring_ would clash with the other dynein monomer in Phi. Our structure of a monomeric human dynein motor domain bound to LIS1 shows that LIS1 can bind to dynein at site^ring^ and site^stalk^ when the motor domain is in its closed confirmation. Thus, the steric incompatibility that was previously predicted (Htet et al., 2020 and Marzo et al., 2020) would apply to the human system as well.

We now indicate in the legend to Figure 1 that the cryo-EM maps are of human dynein’s motor domain.

4. Line 148-149 "…point mutations at M3199A, N3202A, and R3206A", remove "A" after the residues as you are talking about the position, unless you remove "at".

Thanks for catching this. We removed “at”.

5. Line 155-158 "…. in human LIS1 is homologous to yeast S248; however, in our structure T187 faces away from dynein and is poised to hydrogen bond with K175. To make a separation-of-function mutant in human LIS1, the neighboring residue, R186, which extends towards dynein, may serve as a better mutation candidate in future studies (Figure 3E)." It may be better to add a sentence or two here to say what the predicted effect of such a separation-of-function mutation would be on human dynein activation based on the previous data from the budding yeast.

Our prediction would be that a mutation that blocks LIS1’s ability to interact with human dynein at site^stalk^, but not site^ring^ would be less potent at activating human dynein. Given that this is quite speculative we have decided not to include this in the manuscript.

6. Although I do not think functional tests are essential for publication of this structural paper, it would be nice to test if some mutations (designed according to the new structure) affect LIS1's ability to bind dynein or enhance the speed of dynein motility in vitro. Cell-based tests are even better, as the seperation-of-function mutation may affect dynein-mediated cargo transport but not dynein accumulation at the microtubule plus end (Splinter et al., 2012). However, functional assays in humans cells are much harder than those in budding yeast, and some functional tests may need to be done after the publication of this new structure.

We absolutely agree with the reviewer that functional tests would be nice to validate our structural data but, as was pointed out, tests in human cells will be much harder than nuclear segregation or dynein localization assays in yeast and would take much longer than the timeline of a revision would allow.

Reviewer #2 (Recommendations for the authors):I recommend determining whether the K147A mutation in fact disrupts the interaction between b-propellars in human LIS1 and if that prevents dynein stimulation.

We agree with the reviewer that functional validation of the predicted mutations would be desirable. However, as the reviewer points out, this would require significant additional work that would go beyond the timeline of a revision. As Reviewer #1 indicated, experiments in human cells are difficult (and much more time-consuming than the nuclear segregation assays we have used in yeast in the past).